# Visuo-emotional perception and Human Cognition to engineer content-generation using Generative AI

## Abstract

Media platforms compete for users' attention. Their reach crucially depends on algorithmic real-time bidding and efficiency of hyper-personalised, rapidly generated, and user-optimised content. Attention is, however, a scare and fleeting quantity, often awarded less than 1 second per stimulus. Thus, the current strategy is to rely on the vast amount of user-generated data to mimic the content to the user. The underlying assumption is that this is sufficient incentive for attention. This strategy has evidently failed, as witnessed by the alarmingly low or short-lived successes of campaigns in recent times. This mismatch is exacerbated because most content consumed today is digital, whereas strategies for digital content mimic our past understanding of mass-media. Hence, we formalise a new understanding of communication, specifically for the digital media. We prove that the digital medium demands a new understanding of communication protocols. Hence, we take a first principles approach to the new communication protocol: the neurological representations of communication, specifically, where the communication happens in less than 1 second per stimulus. First, we break down and elaborate on this neurological representation of decision-making. Next, we proffer use of our behavioural communication model for generation and optimization of content creatives. To that end, we elaborate methods for rapid, AI-generated content, increasing the efficiency of visual communication on digital media. Within this exploration we include themes of Hyperpersonalisation and Search-engine optimization. Thus, we find that strategically produced content exhibits stronger associations to users' nonconscious needs, wants and goals, which elicits user attention and content-diversity significantly.

## 1 Introduction

In the book Subprime Attention Crisis, Tim Hwang highlights the ineffectiveness of digital advertising. He calls digital advertising "the beating heart of the Internet", but one that is about to collapse. The author compares the Click-Through Rates (CTRs) of the first digital ad that was released in 1994 to the click-through rates on digital advertising in 2018. The famous "You Will" ad (Figure. 1) was released on the popular tech-blog hotwired.com (now Wired.com). Over the next two months, the ad garnered a CTR of 44%, i.e., one in two people who saw the ad, clicked on it. A CTR of 44% is unheard of today when the average for display ads lingers at sub-1% (0.48% in 2018, 0.35% in 2022). Hence, digital communication today is over a 100-times less effective than it was 20 years ago (Marino, 2023; Hwang, 2020).

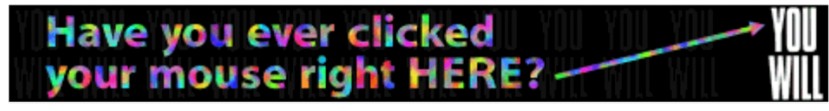

First ad seen on hotwired.com

Figure 1: First digital ad in the world.

The ineffectiveness warrants technological interventions such as:

- Real-Time Bidding,
- Ad Personalation or Targeted Ads,
- Search-Engine, and
- Dynamic Content Optimation

## 1.1 PAST METHODS OF PERSONALISATION

To improve the effectiveness of digital ads, in 2014, Google deployed real-time bidding for its ad spaces. Real-time bidding allows advertisers to manage and optimise ads from multiple ad-networks, enabling them to create and launch campaigns via an instantaneous programmatic auction. In other words, the viewer is exposed to "the right product, at the right time". Search engine optimisation also aims to optimise online content by keywords. The principle of SEO is that there is direct, increasing correlation between the number of keywords in the content and the probability that the content will be clicked-on and consumed. A third method that is used to present "the right content at the right time" is personalisation. Personalisation in communication aims to make communication relevant to the user by incorporating elements of the user's identity into the content or creatives. With personalised ads, also called targeted ads, consumer insights increase the relevancy of the ads presented to the consumer (Meta).

While the war for users' attention wages on delivering precisely and accurately, CTRs continue to decline, from 0.48% in 2018 to 0.35% in 2022. The average Cost per Click (CPC) and Cost per Impression (CPI) of display ads continue to rise, hence the conclusion that online ads mostly fail to work (Marino, 2023).

## 1.2 THE ADOPTION OF GENERATIVE AI

Generative AI methods are evolving rapidly and using text-to-image diffusion models is making it possible to generate images simply based on text input (Ramesh et al., 2022; Rombach et al., 2022; Saharia et al., 2022). This alone has shown incredible promise in scaling up the effort to create visual content at incredible speeds.

However, one of the fallacies of technological advancement is there is often uncertainty in adopting it for large-scale use. Most recently we saw this with the development of the COVID vaccine. Discovery and scaling production of the COVID-19 vaccines was done in record time (Boyle, 2021). But there was still significant hesitation globally to "take the shot". When we observe the widespread adoption of simpler technologies like the hesitance linked to the recent COVID-19 vaccination, or the problem of open defecation in rural India despite the access to clean toilets, it warrants anticipation that the adoption of AI-based Image Generation should go beyond experimental and recreational use and a short-lived hype (Boyle, 2021; Sharma, 2021).

Technology giants such as Google and Microsoft are also not immune to this, having dozens of product failures under the name of Google Graveyard and Microsoft Morgue, respectively (Google; McCracken, 2012).

In this paper, we marry Text-To-Image generation with the centuries-old practice of visual storytelling to create truly compelling creatives. Indeed, one of the first areas of research was visual perception and how the brain processes and represents visuals, even those generated by AI (Churchland et al., 1994).

We look at how to utilise Text-To-Image generation as an effective storytelling tool with the significant advantage of using Gen AI to illustrate stories. Algorithmic personalisation such as dynamic-content optimisation and contextual multi-arm bandit approaches strengthen the prospect of leveraging Generative AI at an unprecedented scale.

## 1.3 MICROSTIMULI FOR THE DIGITAL MEDIUM

In addition to the art and science of storytelling, most communication today is through the digital medium, specifically smartphones. The human brain has evolved over 600 million years. Hence, it

makes sense that the human brain most often resorts to perceptual behaviours stemming from how our early hominid ancestors interacted with their environment (Bennett, 2021). Radio, Television, the Internet, Smartphones, in fact, most modern technology, and what we classify as digital media, emerged only in the last 100 years.

Since the launch of the smartphone in 2007, it has been the spearheading medium of the new digital revolution. As of 2023, there are recorded estimates of 6.84 billion smartphones being used around the world (Howarth, 2023). The global smartphone penetration rates are estimated at 67% with an annual growth rate of 10% per year (Howarth, 2023). With the sheer omnipresence of the smartphone, it has become the final mile for all digital interactions. Hence, this paper focuses on the use of GenAI through the lens of the digital medium of smartphones. Recent studies have come to show that the context-duration on digital mediums is fleeting. Similar studies by Netflix and Facebook have shown the duration of digital decisions to be 1.7 seconds and 1.8 seconds, respectively (FacebookIQ; Netflix).

To solve the problem of the ineffectiveness of creative content in the digital medium, this paper investigates facets of the neuroscience of decision-making, in the (digital) medium. Understanding the processes in the brain gives us insights into what information is crucial and for firing which decision process in the brain.

Identifying these constituents would allow us to design appropriate stimuli that might create the desired response. We propose that this integration of neuroscience and Generative artificial intelligence can help influence smarter, better decisions, such as the adoption of important behaviours such as medical adherence, physical inactivity, and night brushing. In addition, we see applications of the proposed schema in digital marketing of simpler goods and services by increasing CTRs and conversions: the specific areas of focus in this paper.

## 1.4 TO SOLVE THE PROBLEM WE START WITH THE MEDIUM

Studies conducted across different demographics and age groups show an upward trend in the time spent using a smartphone every single day. Earlier studies found that people use their phones on average for about 3 hours and 36 minutes (Anonymous, 2023).

But while the time spent on our phones increases, there is a key interaction metric: the touches on the smartphone screen. In 2016, a study reported that people touch their phones 2617 times in a day (dsc). A more recent study, in 2023, found the new number of touches per day to be 4513 (Anonymous, 2023). The increased frequency of touches hypothesizes a dramatic drop in the time for which any content stays on the screen, where each editorial content on the screen, is in front of the user for a very short duration. To understand the editorial context duration of the smartphone medium, this study recorded the touch interval data from smartphone users. The data indicated that 90% of consecutive touches happened in less than 5 seconds and 95% happened in less than 10 seconds (Anonymous, 2023). This indicates that the editorial context duration of the smartphone is only 5-10 seconds. In comparison, the context duration of the television medium is 5-21 minutes (Cha et al., 2008). For television, a 30-second commercials fit into the subset of the large context duration. However, on smartphones, with a limited context duration window of 5-10 seconds, we need a persuasion stimulus that creates an impact in milliseconds, a MicroStimuli (Anonymous, 2023).

While the concept of MicroStimuli operating in milliseconds on smartphones may seem novel to the conscious mind, it is actually a prevalent aspect of Nature that has been evolving for billions of years. The pioneering work of Nobel laureate Nikolaas Tinbergen has illuminated the concept of Supernormal Stimuli (Tinbergen, 2020). Tinbergen's research revealed how these stimuli possess the remarkable ability to induce fixed action patterns in organisms within mere milliseconds. For instance, male stickleback fish exhibit aggressive behaviours toward the red coloration of a competing male's underbelly during the breeding season. When presented with a red object featuring even more intense red coloration and exaggerated proportions for an extended period, the male sticklebacks displayed heightened aggression, sometimes to the point of exhaustion. This phenomenon in stickleback fish behaviour, where exaggerated and prolonged stimuli trigger heightened responses, offers intriguing parallels to human behaviour and the concept of Fixed Action Patterns: instinctual, pre-programmed sequences of actions triggered by specific stimuli. These patterns are deeply

ingrained in our biology and psychology, and they can be observed in various aspects of human behaviour, from social interactions to decision-making processes.

To provide insight into the time frames associated with decision-making, we delve into the comparison between reflexes and reactions within our bodies as shown in Fig. 2. Human reflexes are remarkably rapid, transpiring in approximately 50 milliseconds, and are managed solely by the spinal cord, bypassing the brain entirely (ref). In the domain of cricket, a batsman necessitates roughly 450 milliseconds to determine the appropriate shot (McLeod, 1987). Additionally, MicroStimuli, which we have meticulously developed to aid in purchase decisions, operate within 920 milliseconds (Anonymous, 2023). Selecting a series or film on Netflix requires a decision time of 1.8 seconds (Netflix). Allocating a donation of USD$1000 to an NGO entails a more reflective 2.5 seconds (Maoz et al., 2019). These diverse scenarios elucidate that, in terms of time, MicroStimuli are ubiquitous and intrinsic components of nature.

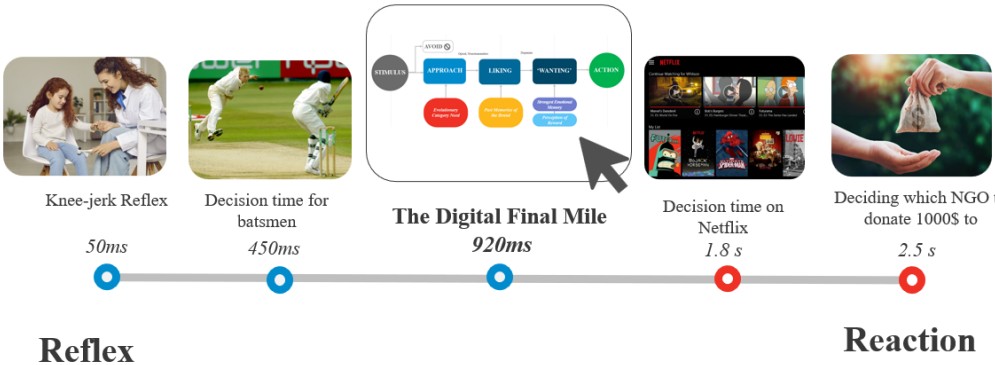

Figure 2: Range of reaction times from reflexes to reactions in the human body.

All long-duration decisions are a combination of numerous micro decisions. The evidence of this statement is illustrated by multiple studies across sports and neuroscience. For instance, a batter dedicates 220ms to discern the incoming ball's trajectory, a mere 10ms for shot determination, and a further 220ms to execute the shot (McLeod, 1987). This indicates that the batter's response to the ball's movement and the subsequent shot selection act as micro-decisions within the broader context of playing the shot. In the realm of neuroscience, findings suggest that the perceived value of a potential outcome is synthesised from its foundational attributes (O'Doherty et al., 2021).

## 2  FIRST PRINCIPLES APPROACH

In the rapidly evolving landscape of e-commerce, understanding the intricacies of consumer decision-making is crucial. To address the challenge of low Click Through Rates (CTRs) and enhance the effectiveness of e-commerce persuasion, we propose the Final Second Framework. At its core, our framework adopts a first principles approach, seeking to uncover the fundamental building blocks of the decision-making process in the final seconds before a purchase decision is made. The product tile is the most repeated stimuli across the e-commerce consumer journey. Displayed on the search page, product display page and even the cart, the product tile, often a simple image against a white background, represents the final mile of persuasion in the buyer's journey. To comprehend the rapidity of this stage, we delve into the neuroscience of visual processing.

It has been discovered that the brain's initial response to any stimulus is to either approach or avoid it, a process that occurs within 350 milliseconds (LeDoux & Bemporad, 1997; Carter, 2019). At this stage, the buyer decides whether the product amongst all the others deserves any attention or not. Following this, the next step in decision-making is the feeling of Liking, which is influenced by a buyer's positive past experiences with a product and occurs within the next 100 milliseconds, at 450 milliseconds (Berridge & Robinson, 2016). ). In the Liking stage, the consideration set of the category is created by the buyer. At the final stage, Liking alone doesn't motivate us to act, hence,

the 'Wanting' stage drives the 'buy now' or 'add to cart' action, this occurs within the next 470 milliseconds (Berridge, 1999; Braeutigam et al., 2001). Totally, the entire decision-making process for an e-commerce purchase can transpire in just 920 milliseconds (Anonymous, 2023).

## 3   New Communication Protocol

To effectively implement the Final Second Framework, we introduce a novel communication protocol that explains how these brain processes can be triggered. First, we evoke an approach response by playing up the evolutionary category need. Categorisation represents the brain's initial phase when assessing any stimulus (Murphy). ). It's noteworthy that a significant 69% of purchase decisions on popular shopping platforms like Amazon.com commence with a category search (Szahun & Dalton). Based on these findings, we suggest that stimulating the evolutionary category need of the product is an ideal way to facilitate the approach response in the buyer's brain. Subsequently, we attain the feeling of Liking by invoking memories associated with the brand (Berridge & O'Doherty, 2014). Therefore, employing brand logos, adhering to brand guidelines, and utilising brand colors serve as cues to foster this sense of Liking in the hedonic centers of the brain. Finally, 'Wanting' is generated through incentive salience. Brain structures (i.e., the mesolimbic system associated with incentive salience) fuelled by dopamine come into play to create 'Wanting'. This incentive salience can be accomplished by activating the strongest emotional memory tied to the brand and enhancing the perception of reward (Anonymous, 2022). The entire decision process is illustrated in Fig. 3.

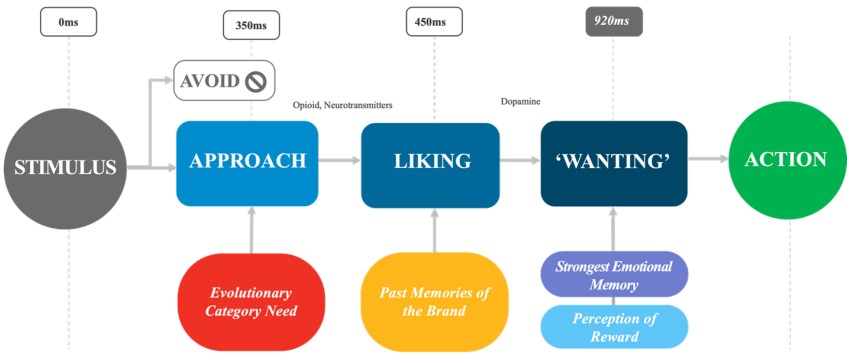

Figure 3: Final Second Framework

## 4   Creating MicroStimuli at Scale using Generative AI

With the advent of diffusion models for image generation, there is a tremendous opportunity to leverage this technology to create personalised creatives at scale, enhance workplace productivity for creative roles, and transform digital marketing. To make this technology a standard in the industry, businesses and brands must adopt it. However, to use these models effectively for businesses, certain guardrails need to be in place to ensure the generation of consistent and meaningful images tailored for smartphones, the primary interface for human-machine interactions. In the traditional marketing industry, creative briefs provided to ad agencies have been crucial for over a century. These briefs define the objectives for creatives, enabling skilled designers to consistently create persuasive stimuli that aligned with the brand strategy. To fully integrate generative AI into marketing, it's essential to provide effective guidelines for AI systems to generate creatives. Prompt engineering, a skill typically possessed by creative individuals, is vital in unlocking the full potential of generative AI in the digital communication domain. Despite technological advancements, the utilisation of generative AI in the digital marketing community remains limited. Therefore, we propose implementing a prompting strategy that bridges the gap between communication strategy and AI-generated creatives. This prompt strategy is rooted in the neuroscience of visual stimulus processing and aims to expedite the creation of end creative materials while meeting the demand for efficient communication in the digital world. The proposed prompt strategy serves a dual purpose. First, it enables machines to consistently generate visually appealing creatives. Second, and most

importantly, it revolutionises marketing practices and enhances customer engagement by creating MicroStimuli—stimuli that trigger decision-making processes in the human brain, leading to action.

Constructs of Prompt Strategy:

1. *Evolutionary category need:*

   When constructing a prompt for generating images from text, it's important to highlight the subject alongside the complementary and relevant style (Liu & Chilton, 2022)). Emphasizing the evolutionary category need of the subject heightens the processing of the visual and creates vivid imagery. For example, the core category need of soft drinks is its ability to quench thirst, this can be visually represented by condensed water droplets on the bottle.

2. *Past memories associated with the brand:* It is crucial to include well-defined brand guidelines, encompassing aspects such as brand color schemes, brand logos and product specifications, in the prompt. These repeated motifs deeply tied to the image of the brand evoke past memories associated with the brand. This allows brands to maintain consistency in representation, while enabling the subjective feeling of pleasure, in familiar users.

3. *Strongest Emotional Memory :* Vivid and emotionally compelling imagery can be attributed to incentive salience and is believed to potentially exert motivational influences. So explicitly calling out the emotional high in the prompt is recommended

4. *Context with photographic details :* In order to elevate the caliber and aesthetics of images generated by generative AI, it is imperative to incorporate additional prompt modifiers. This entails offering comprehensive contextual information for photography and clearly defining the desired image environment. Numerous remarkable visual outcomes created by generative AI and shared online have been attributed to the inclusion of pertinent artistic and photographic directives within the prompt (McCue, 2023).

In Fig. 4, we describe the proof of concept for our prompt strategy. In Fig. 4 (a), the designs are generated by a prompt by ChatGPT for a digital coke ad: Design a compelling digital advertisement for Coca-Cola that demonstrates the brand's enduring appeal and widespread popularity. Highlight the iconic Coca-Cola logo and imagery in a way that seamlessly blends nostalgia with modernity. Incorporate dynamic visuals, catchy music, and a sense of global unity to convey the message that Coca-Cola is a universally recognised symbol of refreshment and enjoyment. Your ad should evoke a feeling of timelessness and cultural resonance, emphasising how Coca-Cola has remained a beloved choice for generations and continues to bring people together in a digital age. In Fig. 4. (b) we used our prompt strategy to create a MicroStimuli: Evolutionary Category Need: a coke bottle with effervescence and water droplets, Past Memories associated with the brand: Product photography, $F40009$ Strongest Emotional Memory: Person drinking coke sitting on a beach mat, looking refreshed, Context with photographic details: view of the beach and people playing volleyball in the background, natural lighting, center composition, — ar $1 : 1$. In Fig. 4. (c) we overlay the brand assets over the generated image. From Fig. 4. We can observe significant leaps in the generative AI creative with the prompt strategy.

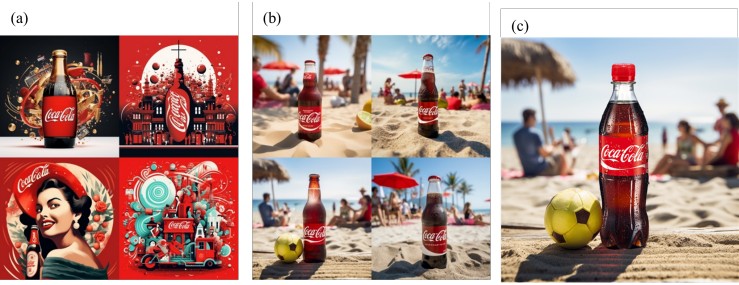

Figure 4: Proof of concept for prompt strategy. (a) prompt generated by chatGPT for a digital coke ad, (b) prompt strategy to create MicroStimuli and (c) brand asset overlayed on the generated image.

## 5 EXPERIMENTAL SETUP

In this research, a Pairwise Comparison (PC) test was executed, in which an online form was developed to simulate a purchasing scenario. The form was segmented into four principal sections:

1. The initial section required subjects to enter their name, email ID, age, and city of residence.
2. The subsequent section established an E-commerce context, prompting subjects to place an item in their cart.
3. In the third segment, subjects were shown an image of the product against a white background along with a designer-created creative for a disinfectant, utilising the Final Second Framework. The specific creatives deployed are depicted in Fig.5.
4. The final section displayed an image of the product, set against a white background, accompanied by a creative for a cold drink. This creative was generated using a prompt strategy built on the Final Second Framework and underwent minimal alterations by a designer to align with brand guidelines. The creatives utilised are illustrated in Fig.6.

The study was centered around two core research inquiries:

1. To assess the impact of the Final Second Framework on the brain's decision-making mechanism.
2. To analyze the efficacy of the prompt strategy built on the Final Second Framework employed for designing a creative through Generative AI.

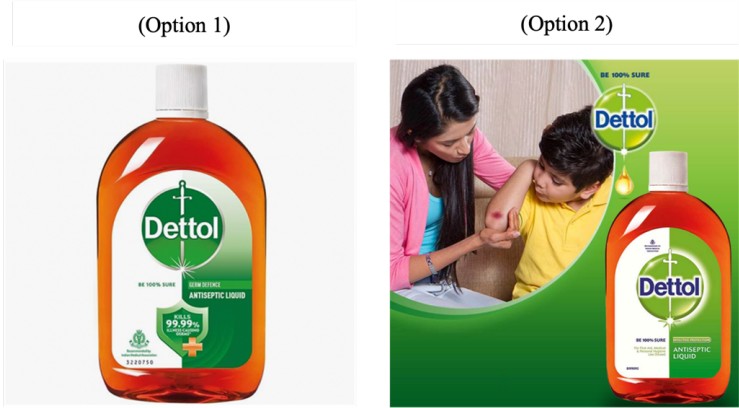

Figure 5: Creatives presented to the subjects for choosing a disinfectant. (Option 1) Product tile of a disinfectant (Option 2) Creative for a disinfectant created by a designer using the Final Second Framework.

## 6 RESULTS AND ANALYSIS

Our randomised sample of subject consisted of a total of 236 individuals who filled the online form. The distribution of age among the subjects is illustrated in Fig. 7, revealing a median age of 27 and an age range spanning from 21 to 55. It is also noted that the majority of the subjects are digital natives, predominantly falling within the 23-35 age bracket.

The demographic data described in Fig. 8 represents the geographical distribution of participants from various countries. Most participants are from India, accounting for 216 individuals, indicating a significant representation from this region. Following India, the United States has the next highest representation with 5 participants. The United Kingdom is represented by 3 participants, and Canada has 2 participants. Australia, France, Italia, Mexico, Ukraine, and the United Arab Emirates each have 1 participant, showing minimal representation from these countries in the study. This distribution illustrates a predominantly Indian demographic, with sparse representation from other regions around the globe.

(Option 1)                    (Option 2)

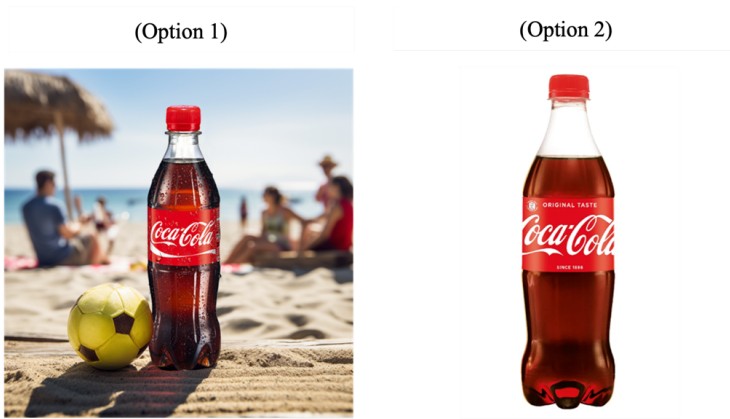

Figure 6: Creatives presented to the subjects for choosing a cold drink. (Option 1) Creative for a cold drink generated by generative AI using the Final Second Framework (Option 2) Product tile of a cold drink.

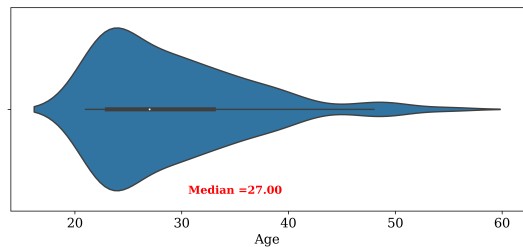

Figure 7: Distribution of age of the subjects.

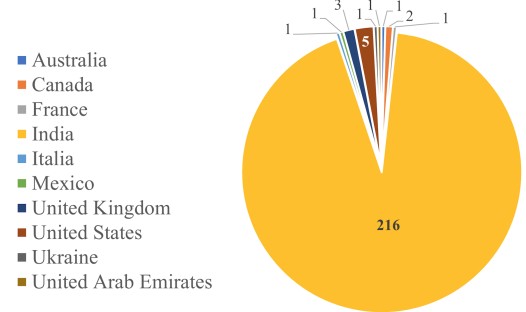

Figure 8: Distribution of country of residence of subjects.

In the study conducted, subjects were given a choice between two distinct options in separate scenarios. In the first scenario, the subjects made selections between the present product tile and the Final Seconds Framework, designer. The results indicated a preference for the Final Seconds Framework, used by the designer with 127 selections, compared to the present product tile, which received 109 selections. In summary, we observe a significant 4% difference between the present product tile and creative generated using the Final Second Framework's prompt strategy. The pie chart describing the results is shown in Fig. 9 (a).

In the second scenario, the subjects chose between the present product tile and the Final Seconds Framework, built using a prompt strategy for Generative AI. In this instance, the Final Seconds Framework with Generative AI was favored, garnering 138 selections, whereas the present product

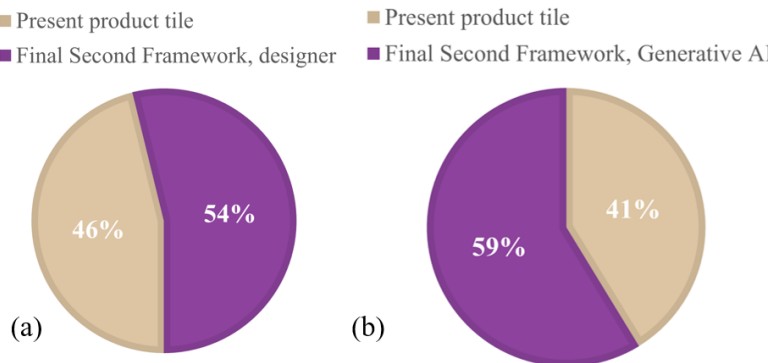

Figure 9: (a) Results of pairwise comparison test on the present product tile and Final Seconds Framework creative built by a designer. (b) Results of pairwise comparison test on the present product tile and Final Seconds Framework creative built using Generative AI.

tile was chosen 97 times. In summary, we observe a significant 9% difference between the present product tile and creative generated using the Final Second Framework's prompt strategy. The pie chart describing the results is shown in Fig. 9 (b). These results suggest a notable inclination of subjects towards the Final Seconds Framework options in both scenarios, highlighting a potential interest or perceived value in these choices over the present product tile.

The outcomes of the conducted study illustrate the prevailing preference for the Final Second Framework over the present product tile in decision-making scenarios. The Final Second Framework, embodying a harmonious integration of Neuroscience and Artificial Intelligence, evidently stands out as a revolutionary approach, contributing significantly to the empowerment of every decision made by humans. The discernible inclination of subjects towards options provided by the Final Second Framework in both scenarios not only underscores its perceived value but also attests to its impactful role in shaping choices.

## 7 SUMMARY AND CONCLUSION

The aim of our study was to investigate whether neuroscientifically designed content (tailored to the final seconds of decision-making) significantly elicits user attention and can be scaled using generative AI. To build our final second framework, we used the first principles approach to better understand the final seconds of an e-commerce purchase. The consumer begins by either Approaching or Avoiding the stimulus, which we evoke by highlighting the core category need of the product. Next, the feeling of Liking, which we further emphasise by invoking memories of the product or similar products using the brand guidelines. Finally, the user ideally reaches the Wanting phase, which leads to Action (purchase). Building on this framework, we used these principles to build a structure for prompt strategy: 1) The evolutionary category need, which heightens the processing of the visual and creates vivid imagery. 2) Past memories, brand guidelines and important motifs the brand uses to maintain consistency. 3) Strongest emotional memory, where making clear the emotional high of the product motivates the consumer to want it. 4) Context with photographic details, where prompt modifiers and quality boosters define the style, dimensions and overall visual appeal of the image. In this study, we applied this to a simulation of purchase from an e-commerce platform. As discussed in the results section, the outcome of the experiments indeed confirmed our hypothesis that the MicroStimuli-centric option is preferred by the user. The two scenarios presented to the user indicated a preference for the MicroStimuli design by a considerable percentage. These results therefore met our expectations and supported our hypotheses. The findings are very relevant to the future progress of e-commerce, as the high spike in phone interactions has compromised our attention spans, and understanding how to accommodate these new conditions is imperative for future successful e-commerce campaigns.

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

## A APPENDIX

### A.1 DISCUSSION

A multitude of studies have been conducted to better understand the neuroscience of decision-making and emphasise the importance of neuromarketing. Smidts (2002) first defined neuromarketing as the study of brain mechanisms and processes to understand consumer behaviour patterns in order to improve marketing strategies. Research on both the neuromarketing methods themselves, as well as consumer behaviour have since been conducted to better understand how marketing can be improved to appeal to consumers on a biological level rather than from only a design perspective. Extensive work detailing and reviewing current findings on neuromarketing methods indicate that neuromarketing has become an essential part of the advertising industry, and methods such as electroencephalograms (EEGs), functional magnetic resonance imaging (fMRIs) and the functional near-infrared spectroscopy (fNIRS) aid the process of understanding consumer behaviour better (Alsharif et al., 2023; Cherubino et al., 2019). Apart from studying the neuroimaging of consumers, research has also been conducted on the role of emotion in affecting these results. Consumers who stated they felt joy, anger or surprise from advertisements displayed different levels of autonomic nervous system activity, heart rate and memory, indicating that different emotions impact an advertisement's effect and must be experimented with further to understand the depth to which this holds true (Baraybar-Fernández et al., 2017). In addition to this, physiological features such as eye movement have been found to be an indicator of what parts of an online advertisement tend to appeal to a consumer, and what results in holding attention for longer (for example, including a picture of a celebrity) (Muñoz-Leiva et al., 2019). Although a solid foundation of research has been established in the field of neuromarketing, the combination of advertisement and artificial intelligence remains relatively nascent. Haleem et al. (2022) critically examines the role of AI in marketing by reviewing the algorithms used to gain consumer insights and optimise data collection and analysis. AI will allow advertisers to identify and track behavioural patterns among consumers, improve personalisation, and utilise trends and other data to choose the optimal time, place, and form of advertisement for the user (Haleem et al., 2022). Similarly, (Nazir et al., 2023) conclude that AI's insights and engagement on social media considerably boosts customer satisfaction resulting in a higher likelihood of a customer repurchasing. These papers review the applications of AI quite thoroughly; however, they do not look at how generative AI could be used as a medium for design itself, through AI-created advertisements. To generate visually pleasing AI generated artwork, Liu and Chilton's prompt engineering guidelines provide a strong foundation. Similarly, Oppenlaeder's taxonomy of prompt modifiers provides valuable insight into how to construct a strategic prompt to yield the best results using style modifiers, quality boosters and 'magic terms' (Oppenlaender, 2022). Pavlichenko & Ustalov (2023) use keywords to find the optimal set of words to generate aesthetically pleasing

outputs, resulting in a vast list of potential high-quality prompt modifiers. While this research helps us approach prompt engineering from a strategic design perspective, these guidelines are tailored towards art and design rather than advertisement, which is what makes this experiment a significant contribution to the field. This research lies in the intersection between neuromarketing and generative artificial intelligence in advertising- a field that has for the most part remained unexplored. Our study provides an investigation into how the two can be used to cater effectively to an audience whose attention spans are rapidly declining on the medium of the future: smartphones. The research does not dispute or challenge the existing theories; rather it integrates neuroscience and AI into an interdisciplinary field which will have lucrative practical implications in the future.

