# OpenReview forum: "Visuo-emotional perception and Human Cognition to engineer content-generation using Generative AI"
_ICLR.cc/2024/Conference — Submitted to ICLR 2024_

### Official Review · Reviewer_Bg7M · 2023-10-16

**Soundness:** 1 poor
**Presentation:** 2 fair
**Contribution:** 1 poor
**Rating:** 3
**Confidence:** 3

**Summary:**

This paper leverages the content generated by generative AI to enhance the critical last moments of decision-making. Grounded in the understanding that long-duration decisions are the cumulative result of numerous micro-decisions, the author dissects the final seconds of an e-commerce purchase into multiple stimuli. Subsequently, the author introduces a four-point prompt strategy, informed by the outcomes of this analysis. After that, to validate the efficacy of this prompt strategy, the author conducted a series of experiments.

**Strengths:**

Thanks for your interests in ICLR! Overall, this is an interesting paper on a topic which is of interest to ICLR Conference. It offers valuable insights into the application of neuroscientifically designed content to enhance ad click-through rates. The paper astutely recognizes the promise of leveraging Generative AI for this purpose. Building upon this foundation, the author thoughtfully presents four distinct prompt strategies and supports them with well-structured experiments, thus substantiating the validity of their approach.

**Weaknesses:**

While the author presents a comprehensive theoretical framework and provides clear and detailed insights into the prompt strategies, there is room for improvement in the experimental validation of the proposed techniques. As outlined in the paper, the experiments are limited to a single product tested on a sample of 236 participants. Given the potential applicability of this technique to a wide range of products, the scope of experimentation appears somewhat narrow. Expanding the experiment set to encompass a more diverse array of products would strengthen the paper's claims.

Additionally, the manual design of prompts by the author may not be a scalable solution when considering the need for ad design across a vast array of products. Further exploration of automated or semi-automated prompt generation methods could enhance the paper's practicality and applicability in real-world scenarios.

**Questions:**

1. Have you explored the possibility of automating the prompt generation process for various products?

2. Could you provide insights into any supplementary experiments conducted to further validate the effectiveness of the prompt strategy outlined in your paper?

---

### Official Review · Reviewer_DN2b · 2023-10-19

**Soundness:** 3 good
**Presentation:** 3 good
**Contribution:** 2 fair
**Rating:** 3
**Confidence:** 4

**Summary:**

The paper proposes a framework that leverages Generative AI to create Ad creatives that aim to increase the Click-through rate of advertisements. The framework and ad creative generation leverage four principles: 1) the evolutionary category need, 2) past memories and brand guidelines; 3) the strongest emotional memory, and 4) context with photographic details.

**Strengths:**

The paper does a good job of motivating and explaining the problem, as well as providing all the necessary details and motivation to understand the necessary background. Also, the paper focuses on an interesting aspect of generative AI and how it can be used to generate ad creatives with the goal of increasing click-through rates that, over the years, have been declining. Overall, I think that this work has the potential to inform various interested stakeholders, including advertisers, policymakers, and social media operators. Also, I like the paper’s approach that aims to leverage the power of Generative AI (particularly ChatGPT) to generate content based on principles obtained from the neuroscience field.

**Weaknesses:**

My main concerns with the paper are related to the framework’s evaluation. I believe that the evaluation is quite limited and simplistic, given that the sample of the recruited participants is biased (the overwhelming majority being from India) and the evaluation focuses on only two products. I suggest that the authors explain and motivate how they perform the user recruitment procedure and the reason why they selected the two products. Overall, given these limitations, it’s unclear whether the paper’s results are generalizable.

Additionally, the paper fails to explain how this study is different from previous efforts that aim to understand the use of neuromarketing methods without the use of Generative AI to create the ad creatives. The presented framework can also be applied by people to generate ad creatives, so its unclear if the novelty of this work lies in the formulation/use of the framework or the combination of the framework with Generative AI models like ChatGPT. I suggest to the authors to better contextualize their work and better explain the novelty of this work.

Also, the paper does not explain how the envisioned framework will be applied in practice. The paper’s evaluation defines a set of prompts that are very specific to the products that are studied and generates creatives that are then subsequently used to compare the user perceptions vs. ad creatives that simply show the product with a white background. Overall, it’s unclear on whether the envisioned framework can be applied in practice without great input and effort from experts that will guide the generation of the ad creatives.

In addition, there is a disconnection between the motivation of the work and the framework/evaluation. The framework does not account for user personalization, which is an important aspect when considering the ad ecosystem. So I am wondering how the paper is planning to incorporate user personalization in this framework and how Generative AI models can assist in this, especially when considering the privacy concerns that may arise from sharing user-specific data with companies that offer LLM solutions (e.g., OpenAI).

To summarize, I believe that this work is interesting and important, however, at this stage, I believe that the paper is not ready for publication. In addition to the above concerns, I would like to make the following suggestions to the authors (mainly minor issues):
1. There are a couple of references listed as Anonymous, when they are not Anonymous so I suggest fixing these issues.
2. Consider not using pie charts for the evaluation results, given that it is one of the worst visualization methods.

**Questions:**

1. How did you recruit participants, and why most of them are from India? How can the recruitment approach affect the presented results?
2. How are the two products selected? Are these products popular in India, where most participants are from?
3. How is this study different from previous efforts studying the use of neuromarketing methods vs. plain advertisements like the ones shown to the participants (plain background with the product in the middle)? Is the novelty of the work the use of ChatGPT to generate the ad creatives?

---

### Official Review · Reviewer_1HkZ · 2023-10-30

**Soundness:** 1 poor
**Presentation:** 3 good
**Contribution:** 2 fair
**Rating:** 1
**Confidence:** 5

**Summary:**

This paper mainly investigates whether generative AI can produce content to attract the user and explains the procedure of content generation from the neuroscience perspective.

**Strengths:**

1. This paper introduces a communication protocol to explain how the brain has been triggered, the entire process is fluent and reasonable.
2. The content of the pre-research is sufficient.
3. The strategy of the prompt is meaningful.

**Weaknesses:**

1. This work is too simple, just using the existing GenAI to produce the context and comparing it with the corresponding items.
2. The prompt is hand-crafted, and cannot be applied flexibly.
3. The number of samples in the experiment is too small, and the experiments should cover more scenarios.

**Questions:**

1. How do you confirm the prompt is reliable and the output of the GenAI is following the rules?
2. Since it's an online experiment, why not invite more people?

---

### Official Review · Reviewer_ZAnW · 2023-10-31

**Soundness:** 2 fair
**Presentation:** 2 fair
**Contribution:** 2 fair
**Rating:** 3
**Confidence:** 2

**Summary:**

The paper proposes a method for rapid, AI-generated content, increasing the efficiency of visual communication on digital media. Within
this exploration the authors include themes of Hyperpersonalisation and Search-engine optimation.

**Strengths:**

I have not found any strengths of this paper.

**Weaknesses:**

1. The theme of this paper may not be closely related to the conference, as it is only an engineering specification and lacks theoretical explanation.

2. The presentation of the paper is chaotic, making it difficult to read.

3. The method mentioned in the paper, which utilizes ChatGTP to generate accurate prompts and generates high-quality digital advertisements using this prompts and a large text-to-image model, has been widely applied in the engineering field and therefore lacks innovation.

**Questions:**

Please refer to Weaknesses.

---

### Meta-Review · Area_Chair_poho · 2023-12-08

**Metareview:**

This paper explores how ads generated with LLMs could lead to higher click-through rate. The strength of the paper includes the problem definition and  the simple approach. The limitations of the paper include 1) lack of technical novelty, 2) limited scalability due to manual prompting, 3) insufficient evaluation. All reviewers are in agreement that the paper needs much more work, and there is no response from the authors.

**Justification For Why Not Higher Score:**

All reviewers recommend rejecting this paper.

**Justification For Why Not Lower Score:**

N/A

---

### Decision · Program_Chairs · 2024-01-16

Reject